# SELECTIVE SELF-TRAINING FOR SEMI-SUPERVISED LEARNING

## ABSTRACT

Semi-supervised learning (SSL) is a study that efficiently exploits a large amount of unlabeled data to improve performance in conditions of limited labeled data. Most of the conventional SSL methods assume that the classes of unlabeled data are included in the set of classes of labeled data. In addition, these methods do not sort out useless unlabeled samples and use all the unlabeled data for learning, which is not suitable for realistic situations. In this paper, we propose an SSL method called *selective self-training* (SST), which selectively decides whether to include each unlabeled sample in the training process. It is also designed to be applied to a more real situation where classes of unlabeled data are different from the ones of the labeled data. For the conventional SSL problems which deal with data where both the labeled and unlabeled samples share the same class categories, the proposed method not only performs comparable to other conventional SSL algorithms but also can be combined with other SSL algorithms. While the conventional methods cannot be applied to the new SSL problems where the separated data do not share the classes, our method does not show any performance degradation even if the classes of unlabeled data are different from those of the labeled data.

## 1 INTRODUCTION

Recently, machine learning has achieved a lot of success in various fields and well-refined datasets are considered to be one of the most important factors (Everingham et al., 2010; Krizhevsky et al., 2012; Russakovsky et al., 2015). Since we cannot discover the underlying real distribution of data, we need a lot of samples to estimate it correctly (Nasrabadi, 2007). However, creating a large amount of dataset requires a huge amount of time, cost and manpower (Odena et al., 2018).

Semi-supervised learning (SSL) is a method relieving the inefficiencies in data collection and annotation process, which lies between the supervised learning and unsupervised learning in that both labeled and unlabeled data are used in the learning process (Chapelle et al., 2009; Odena et al., 2018). It can efficiently learn a model from fewer labeled data using a large amount of unlabeled data (Zhu, 2006). Accordingly, the significance of SSL has been studied extensively in the previous literatures (Zhu et al., 2003; Rosenberg et al., 2005; Kingma et al., 2014; Rasmus et al., 2015; Odena, 2016). These results suggest that SSL can be a useful approach in cases where the amount of annotated data is insufficient.

However, there is a recent research discussing the limitations of conventional SSL methods (Odena et al., 2018). They have pointed out that conventional SSL algorithms are difficult to be applied to real applications. Especially, the conventional methods assume that all the unlabeled data belong to one of the classes of the training labeled data. Training with unlabeled samples whose class distribution is significantly different from that of the labeled data may degrade the performance of traditional SSL methods. Furthermore, whenever a new set of data is available, they should be trained from the scratch using all the data including out-of-class[1] data.

---

[1]The term *out-of-class* is used to denote the situation where the new dataset contains samples originated from different classes than the classes of the old data. On the other hand, the term *in-class* is used when the new data contain only the samples belonging to the previously observed classes.

In this paper, we focus on the classification task and propose a deep neural network based approach named as *selective self-training* (SST) to solve the limitation mentioned above. Unlike the conventional self-training methods in (Chapelle et al., 2009), our algorithm selectively utilizes the unlabeled data for the training. To enable learning to select unlabeled data, we propose a *selection network*, which is based on the deep neural network, that decides whether each sample is to be added or not. Different from (Wang et al., 2018), SST does not use the classification results for the data selection. Also, we adopt an ensemble approach which is similar to the co-training method (Blum & Mitchell, 1998) that utilizes outputs of multiple classifiers to iteratively build a new training dataset. In our case, instead of using multiple classifiers, we apply a temporal ensemble method to the selection network. For each unlabeled instance, two consecutive outputs of the selection network are compared to keep our training data clean. In addition, we have found that the balance between the number of samples per class is quite important for the performance of our network. We suggest a simple heuristics to balance the number of selected samples among the classes. By the proposed selection method, reliable samples can be added to the training set and uncertain samples including out-of-class data can be excluded.

SST is a self-training framework, which iteratively adopts the newly annotated training data (details in Section 2.1). SST is also suitable for the incremental learning which is frequently used in many real applications when we need to handle gradually incoming data. In addition, the proposed SST is suitable for lifelong learning which makes use of more knowledge from previously acquired knowledge (Thrun & Mitchell, 1995; Carlson et al., 2010; Chen & Liu, 2018). Since SSL can be learned with labeled and unlabeled data, any algorithm for SSL may seem appropriate for lifelong learning. However, conventional SSL algorithms are inefficient when out-of-class samples are included in the additional data. SST only add samples having high relevance in-class data and is suitable for lifelong learning. The main contributions of the proposed method can be summarized as follows:

- For the conventional SSL problems, the proposed SST method not only performs comparable to other conventional SSL algorithms but also can be combined with other algorithms.
- For the new SSL problems, the proposed SST does not show any performance degradation even with the out-of-class data.
- SST requires few hyper-parameters and can be easily implemented.
- SST is more suitable for lifelong learning compared to other SSL algorithms.

To prove the effectiveness of our proposed method, first, we conduct experiments comparing the classification errors of SST and several other state-of-the-art SSL methods (Laine & Aila, 2016; Tarvainen & Valpola, 2017; Luo et al., 2017; Miyato et al., 2017) in conventional SSL settings. Second, we propose a new experimental setup to investigate whether our method is more applicable to real-world situations. The experimental setup in (Odena et al., 2018) samples classes among in-classes and out-classes. In the experimental setting in this paper, we sample unlabeled instances evenly in all classes. (details in Section 6.6 of the supplementary material). We evaluate the performance of the proposed SST using three public benchmark datasets: CIFAR-10, CIFAR-100 (Krizhevsky & Hinton, 2009), and SVHN (Netzer et al., 2011).

## 2 BACKGROUND

In this section, we introduce the background of our research. First, we introduce some methods of self-training (McLachlan, 1975; Zhu, 2007; Zhu & Goldberg, 2009) on which our work is based. Then we describe consistency regularization-based algorithms such as temporal ensembling (Laine & Aila, 2016).

### 2.1 SELF-TRAINING

Self-training method has long been used for semi-supervised learning (McLachlan, 1975; Rosenberg et al., 2005; Zhu, 2007; Zhu & Goldberg, 2009). It is a resampling technique that repeatedly labels unlabeled training samples based on the confidence scores and retrains itself with the selected pseudo-annotated data. Our proposed method can also be categorized as a self-training method. Figure 1 shows an overview of our SSL system. Since our proposed algorithm is based on the self-training, we follow its learning process. This process can be formalized as follows. (i) Training a

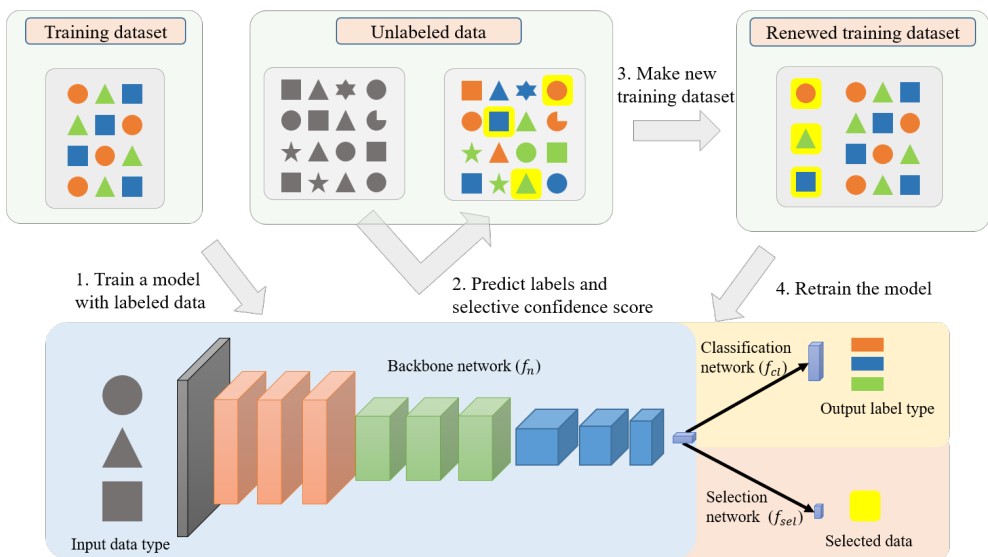

Figure 1: An overview of the proposed SST. Different shapes represent the input data with different underlying distribution, and different colors (orange, blue, and green) are for different classes. In the initial training dataset, only three classes with their corresponding distributions ($\bigcirc$, $\square$, $\triangle$) exist and are used for initial training. Then the unlabeled data which include unseen distribution ($\bigstar$, $\maltese$) are inputted to the classification as well as the selection network. At the bottom right, unlabeled samples with higher selection network output values than a certain threshold are denoted by yellow and selected to be included in the training process for the next iteration, while the remaining are not used for training.

model with labeled data. (ii) Predicting unlabeled data with the learned model. (iii) Retraining the model with labeled and selected pseudo-labeled data. (iv) Repeating the last two steps.

However, most self-training methods assume that the labeled and unlabeled data are generated from the identical distribution. Therefore, in real-world scenarios, some instances with low likelihood according to the distribution of the labeled data are likely to be misclassified inevitably. Consequently, these erroneous samples significantly lead to worse results in the next training step. To alleviate this problem, we adopt the ensemble and balancing methods to select reliable samples.

## 2.2 CONSISTENCY REGULARIZATION

Consistency regularization is one of the popular SSL methods and has been referred to many recent researches (Laine & Aila, 2016; Miyato et al., 2017; Tarvainen & Valpola, 2017). Among them, $\Pi$ model and temporal ensembling are widely used (Laine & Aila, 2016). They have defined new loss functions for unlabeled data. The $\Pi$ model outputs $f(\mathbf{x})$ and $\hat{f}(\mathbf{x})$ for the same input $\mathbf{x}$ by perturbing the input with different random noise and using dropout (Srivastava et al., 2014), and then minimizes the difference ($\|f(\mathbf{x}) - \hat{f}(\mathbf{x})\|^2$) between these output values. Temporal ensembling does not make different predictions $f(\mathbf{x})$ and $\hat{f}(\mathbf{x})$, but minimizes the difference ($\|f_{t-1}(\mathbf{x}) - f_t(\mathbf{x})\|^2$) between the outputs of two consecutive iterations for computational efficiency. In spite of the improvement in performance, they require lots of things to consider for training. These methods have various hyperparameters such as 'ramp up', 'ramp down', 'unsupervised loss weight' and so on. In addition, customized settings for training such as ZCA preprocessing and mean-only batch normalization (Salimans & Kingma, 2016) are also very important aspects for improving the performance (Odena et al., 2018).

---

**Algorithm 1** Training procedure of the proposed SST

---

**Require**: $\mathbf{x}_i, y_i$: training data and label
**Require**: $\mathcal{L}, \mathcal{U}$: labeled and unlabeled datasets
**Require**: $\mathcal{I}_U$: set of unlabeled sample indices
**Require**: $f_n(\cdot; \theta_n)$, $f_{cl}(\cdot; \theta_c)$ and $f_{sel}(\cdot; \theta_s)$: trainable SST model
**Require**: $\alpha, \epsilon, K, K_{re}$: hyper-parameters, $0 \leq \alpha < 1, 0 \leq \epsilon < 1$
 1: randomly initialize $\theta_n, \theta_c, \theta_s$
 2: train $f_n(\cdot; \theta_n)$, $f_{cl}(\cdot; \theta_c)$ and $f_{sel}(\cdot; \theta_s)$ for $K$ epochs using $\mathcal{L}$
 3: **repeat**
 4:     initialize $r_i^t = -1, \mathcal{I}_S = \varnothing$
 5:     **for** each $i \in \mathcal{I}_U$ **do**
 6:         $r_i^{t-1} \leftarrow r_i^t, \quad r_i^t \leftarrow f_{cl}(f_n(\mathbf{x}_i; \theta_n); \theta_c), \quad s_i \leftarrow f_{sel}(f_n(\mathbf{x}_i; \theta_n); \theta_s)$
 7:         **if** $r_i^{t-1} \neq r_i^t$ **then**
 8:             $z_i \leftarrow 0$
 9:         **end if**
10:         $z_i \leftarrow \alpha z_i + (1 - \alpha) s_i$
11:         **if** $z_i > 1 - \epsilon$ **then**
12:             $\mathcal{I}_S \leftarrow \mathcal{I}_S \cup \{i\}$
13:             assign label for $\mathbf{x}_i$ using $r_i$
14:         **end if**
15:     **end for**
16:     update $\mathcal{U}_S$ with data balancing
17:     $\mathcal{T} \leftarrow \mathcal{L} \cup \mathcal{U}_S$
18:     retrain $f_n(\cdot; \theta_n)$, $f_{cl}(\cdot; \theta_c)$ and $f_{sel}(\cdot; \theta_s)$ for $K_{re}$ epochs using $\mathcal{T}$
19: **until** stopping criterion is *true*

---

## 3 METHOD

In this section, we introduce our selective self-training (SST) method. The proposed model consists of three networks as shown in the bottom part of Figure 1. The output of the backbone network is fed into two sibling fully-connected layers —a classification network $f_{cl}(\cdot; \theta_c)$ and a selection network $f_{sel}(\cdot; \theta_s)$, where $\theta_c$ and $\theta_s$ are learnable parameters for each of them. In this paper, we define the classification result and the selection score as $r_i = f_{cl}(f_n(x_i; \theta_n); \theta_c)$ and $s_i = f_{sel}(f_n(x_i; \theta_n); \theta_s)$, respectively, where $f_n(\cdot; \theta_n)$ denotes the backbone network with learnable parameters $\theta_n$. Note that we define $r_i$ as the resultant label and it belongs to one of the class labels $r_i \in \mathcal{Y} = \{1, 2, \cdots, C\}$. The network architecture of the proposed model is detailed in Section 6.2 in the supplementary material. As shown in Figure 1, the proposed SST method can be represented in the following four steps. First, SST trains the network using a set of the labeled data $\mathcal{L} = \{(\mathbf{x}_i, y_i) \mid i = 1, \cdots, L\}$, where $\mathbf{x}_i$ and $y_i \in \{1, 2, \cdots, C\}$ denote the data and the ground truth label respectively, which is a standard supervised learning method. The next step is to predict all the unlabeled data $\mathcal{U} = \{\mathbf{x}_i \mid i = L + 1, \cdots, N\}$ and select a subset of the unlabeled data $\{\mathbf{x}_i \mid i \in \mathcal{I}_S\}$ whose data have high selection scores with the current trained model, where $\mathcal{I}_S$ denotes a set of selected sample indices from $\mathcal{I}_U = \{L + 1, \cdots, N\}$. Then, we annotate the selected samples with the pseudo-categories evaluated by the classification network and construct a new training dataset $\mathcal{T}$ composed of $\mathcal{L}$ and $\mathcal{U}_S = \{(\mathbf{x}_i, \hat{y}_i) \mid i \in \mathcal{I}_S\}$. After that, we retrain the model with $\mathcal{T}$ and repeat this process iteratively. The overall process of the SST is described in Algorithm 1 and the details of each of the four steps will be described later.

### 3.1 SUPERVISED LEARNING

The SST algorithm first trains a model with supervised learning. At this time, the entire model (all three networks) is trained simultaneously. The classification network is trained using the softmax function and the cross-entropy loss as in the ordinary supervised classification learning task. In case of the selection network, the training labels are motivated by discriminator of generative adversarial networks (GAN) (Goodfellow et al., 2014; Yoo et al., 2017). When $i$-th sample $\mathbf{x}_i$ with the class

label $y_i$ is fed into the network, the target for the selection network is set as:

$$g_i = \begin{cases} 1, & \text{if } r_i = y_i \text{ for } i \in \mathcal{I}_L \\ 0, & \text{if } r_i \neq y_i \text{ for } i \in \mathcal{I}_L \end{cases} \tag{1}$$

where $\mathcal{I}_L = \{1, \cdots, L\}$ represents a set of labeled sample indices. The selection network is trained with the generated target $g_i$. Especially, we use the sigmoid function for the final activation and the binary cross-entropy loss to train the selection network. Our selection network does not utilize the softmax function because it produces a relative value and it can induce a high value even for an out-of-class sample. Instead, our selection network is designed to estimate an absolute confidence score using the sigmoid activation function. Consequently, our final loss function is a sum of the classification loss $L_{cl}$ and the selection loss $L_{sel}$:

$$L_{total} = L_{cl} + L_{sel}. \tag{2}$$

## 3.2 PREDICTION AND SELECTION

After learning the model in a supervised manner, SST takes all instances of the unlabeled set $\mathcal{U}$ as input and predicts classification result $r_i$ and the selection score $s_i$, for all $i \in \mathcal{I}_U$. We utilize the classification result and selection score ($r_i$ and $s_i$) to annotate and choose unlabeled samples, respectively. In the context of self-training, removing erroneously annotated samples is one of the most important things for the new training dataset. Thus, we adopt temporal co-training and ensemble methods for selection score in order to keep our training set from contamination. First, let $r_i^t$ and $r_i^{t-1}$ be the classification results of the current and the previous iterations respectively and we utilize the temporal consistency of these values. If these values are different, we set the ensemble score $z_i = 0$ to reduce uncertainty in selecting unlabeled samples. Second, inspired by (Laine & Aila, 2016), we also utilize multiple previous network evaluations of unlabeled instances by updating the ensemble score $z_i = \alpha z_i + (1 - \alpha)s_i$, where $\alpha$ is a momentum weight for the moving average of ensemble scores. However, the aim of our ensembling approach is different from (Laine & Aila, 2016). They want to alleviate different predictions for the same input, which are resulted from different augmentation and noise to the input. However, our aim differs from theirs in that we are interested in selecting reliable (pseudo-)labeled samples. After that, we select unlabeled samples with high ensemble score $z_i$. It is very important to set an appropriate threshold because it decides the quality of the added unlabeled samples for the next training. If the classification network is trained well on the labeled data, the training accuracy would be very high. Since the selection network is trained with the target $g_i$ generated from the classification score $r_i$, the selection score $s_i$ will be close to 1.0. We set the threshold to $1 - \epsilon$ and control it by changing $\epsilon$. In this case, if the ensemble score $z_i$ exceeds $1 - \epsilon$, the pseudo-label of the unlabeled sample $\hat{y}_i$ is set to the classification result $r_i$.

## 3.3 NEW TRAINING DATASET

When we construct a new training dataset, we keep the number of samples of each class the same. The reason is that if one class dominates the others, the classification performance is degraded by the imbalanced distribution (FernáNdez et al., 2013). We also empirically found that naively creating a new training dataset fails to yield good performance. In order to fairly transfer the selected samples to the new training set, the amount of migration in each class should not exceed the number of the class having the least selected samples. We take arbitrary samples in every class as much as the maximum number satisfying this condition. The new training set $\mathcal{T}$ is composed of both a set of labeled samples $\mathcal{L}$ and a set of selected unlabeled samples $\mathcal{U}_S$. The number of selected unlabeled samples is the same for all classes.

## 3.4 RE-TRAINING

After combining the labeled and selected pseudo-labeled data, the model is retrained with the new dataset for $K_{re}$ epochs. In this step, the label for the selection network is obtained by a process similar to Eq. (1). Above steps (except for Section 3.1) are repeated for $M$ iterations until (near-) convergence.

Table 1: Ablation study with 5 runs on the CIFAR-10 dataset. 'balance' denotes the usage of data balancing scheme during data addition as described in Sec. 3.3, 'ensemble' is for the usage of previous selection scores as in the 10th line of Algorithm 1, and 'multiplication' is the scheme of multiplying top-1 softmax output of the classifier network to the selection score and use it as a new selection score.

| method | balance | ensemble | multiplication | error |
|---|---|---|---|---|
| supervised learning | | | | $18.97 \pm 0.37\%$ |
| SST | x | x | x | $21.44 \pm 4.05\%$ |
| | o | x | x | $14.43 \pm 0.43\%$ |
| | o | o | x | $11.82 \pm 0.40\%$ |
| | o | o | o | $11.86 \pm 0.15\%$ |

## 4 EXPERIMENTS

To evaluate our proposed SST algorithm, we conduct two types of experiments. First, we evaluate the proposed SST algorithm for the conventional SSL problem where all unlabeled data are in-class. Then, SST is evaluated with the new SSL problem where some of the unlabeled data are out-of-class. In the case of in-class data, gradually gathering highly confident samples in $\mathcal{U}$ can help improve the performance. On the other hand, in the case of out-of-class data, a strict threshold is preferred to prevent uncertain out-of-class data from being involved in the new training set. Therefore, we have experimented with *decay* mode that decreases the threshold in log-scale and *fixed* mode that fixes the threshold in the way described in Section 4.2. We have experimented our method with 100 iterations and determined epsilon by cross-validation in *decay* modes. In case of *fixed* modes, epsilon is fixed and the number of iteration is determined by cross-validation. The details about the experimental setup and the network architecture are presented in Section 6.1, 6.2 of the supplementary material.

### 4.1 CONVENTIONAL SSL PROBLEMS WITH IN-CLASS UNLABELED DATA

We experiment with a couple of simple synthetic datasets (two moons, four spins) and three popular datasets which are SVHN, CIFAR-10, and CIFAR-100 (Netzer et al., 2011; Krizhevsky et al., 2014). The settings of labeled versus unlabeled data separation for each dataset are the same with (Laine & Aila, 2016; Miyato et al., 2017; Tarvainen & Valpola, 2017). More details are provided in Section 6.3 in the supplementary material. The experimental results of the synthetic datasets can be found in Section 6.4 of the supplementary material.

#### 4.1.1 ABLATION STUDY

We have performed experiments on CIFAR-10 dataset with the combination of three types of components. As described in Table 1, these are whether to use data balancing scheme described in Section 3.3 (balance), whether to use selection score ensemble in the 10th line of Algorithm 1 (ensemble) and whether to multiply the selection score with the top-1 softmax output of the classifier network to set a new selection score for comparison with the threshold (multiplication). First, when SST does not use all of these, the error 21.44% is higher than that of the supervised learning which does not use any unlabeled data. This is due to the problem of unbalanced data mentioned in subsection 3.3. When the data balance is used, the error is 14.43%, which is better than the baseline 21.44%. Adding the ensemble scheme results in 11.82% error, and the multiplication scheme shows a slight drop in performance. Since all of the experiments use the same threshold, the number of candidate samples to be added is reduced by the multiplication with the top-1 softmax output and the variation becomes smaller because only confident data are added. However, we have not used the multiplication scheme in what follows because the softmax classification output is dominant in multiplication. Therefore, we have used only balance and ensemble schemes in the following experiments.

#### 4.1.2 EXPERIMENTAL RESULTS

Table 2 shows the experiment results of supervised learning, conventional SSL algorithms and the proposed SST on CIFAR-10, SVHN and CIFAR-100 datasets. Our baseline model with supervised learning performs slightly better than what has been reported in other papers (Laine & Aila, 2016; Tarvainen & Valpola, 2017; Luo et al., 2017) because of our different settings such as Gaussian noise

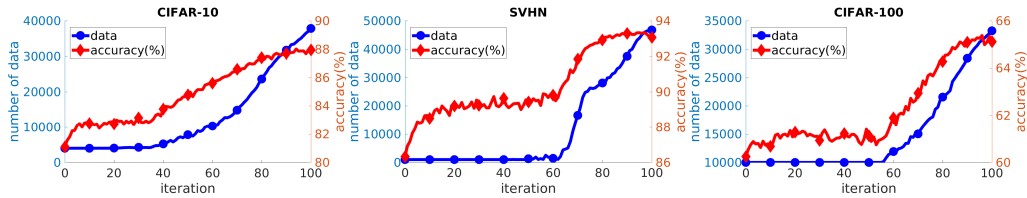

Figure 2: SST result on CIFAR-10, SVHN, and CIFAR-100 datasets with 5 runs. The $x$-axis is the iteration, the blue circle is the average of the number of data used for training, and the red diamond is the average accuracy.

Table 2: Classification error on CIFAR-10 (4k Labels), SVHN (1k Labels), and CIFAR-100 (10k Labels) with 5 runs using in-class unlabeled data (* denotes that the test has been done by ourselves).

| Method | CIFAR-10 | SVHN | CIFAR-100 |
|---|---|---|---|
| Supervised (sampled)* | $18.97 \pm 0.37\%$ | $13.45 \pm 0.92\%$ | $40.24 \pm 0.45\%$ |
| Supervised (all)* | $5.57 \pm 0.07\%$ | $2.87 \pm 0.06\%$ | $23.36 \pm 0.27\%$ |
| Mean Teacher (Tarvainen & Valpola, 2017) | $12.31 \pm 0.28\%$ | $3.95 \pm 0.21\%$ | - |
| Π model (Laine & Aila, 2016) | $12.36 \pm 0.31\%$ | $4.82 \pm 0.17\%$ | $39.19 \pm 0.36\%$ |
| TempEns (Laine & Aila, 2016) | $12.16 \pm 0.24\%$ | $4.42 \pm 0.16\%$ | $38.65 \pm 0.51\%$ |
| TempEns + SNTG (Luo et al., 2017) | $10.93 \pm 0.14\%$ | $3.98 \pm 0.21\%$ | $40.19 \pm 0.51\%*$ |
| VAT (Miyato et al., 2017) | $11.36 \pm 0.34\%$ | $5.42 \pm 0.22\%$ | - |
| VAT + EntMin (Miyato et al., 2017) | $10.55 \pm 0.05\%$ | $3.86 \pm 0.11\%$ | - |
| pseudo-label (Lee, 2013; Odena et al., 2018) | $17.78 \pm 0.57\%$ | $7.62 \pm 0.29\%$ | - |
| Proposed method (SST)* | $11.82 \pm 0.40\%$ | $6.88 \pm 0.59\%$ | $34.89 \pm 0.75\%$ |
| SST + TempEns + SNTG* | $9.99 \pm 0.31\%$ | $4.74 \pm 0.19\%$ | $34.94 \pm 0.54\%$ |

on inputs, optimizer selection, the mean-only batch normalizations and the learning rate parameters. For all the datasets, we have also performed experiments with a model of SST combined with the temporal ensembling (TempEns) and SNTG, labeled as SST+TempEns+SNTG in the table. For the model, the pseudo-labels of SST at the last iteration is considered as the true class label. Figure 2 shows the number of samples used in the training and the corresponding accuracy on the test set for each dataset.

**CIFAR-10:** The baseline network yields the test error of 18.97% and 5.57% when trained with 4,000 (sampled) and 50,000 (all) labeled images respectively. The test error of our SST method reaches 11.82% which is comparable to other algorithms while SST+TempEns+SNTG model results 1.83% better than the SST-only model.

**SVHN:** The baseline model for SVHN dataset is trained with 1,000 labeled images and yields the test error of 13.45%. Our proposed method has an error of 6.88% which is relatively higher than those of other SSL algorithms. Performing better than SST, SST+TempEns+SNTG reaches 4.74% of error which is worse than that of TempEns+SNTG model. We suspect two reasons for this. The first is that SVHN dataset is not well balanced, and the second is that SVHN is a relatively easy dataset, so it seems to be easily added to the hard labels. With data balancing, the SST is still worse than other algorithms. More details are provided in Section 6.5 in the supplementary material. We think this phenomenon owes to the use of hard labels in SST where incorrectly estimated samples deteriorate the performance.

**CIFAR-100 :** While the baseline model results in 40.24% of error rate through supervised learning with sampled data, our method performs with 34.89% of error, enhancing the performance by 5.3%. We have observed that the performance of TempEns+SNTG is lower than TempEns, and when TempEns+SNTG is added to SST, performance is degraded slightly. Although TempEng+SNTG shows better performance than TempEng without augmentation in (Luo et al., 2017), its performance is worse than that of TempEng with augmentation in our experiment. [2]. The reason for this can be

---

[2]We have reproduced tempEns+SNTG model with a Pytorch implementation, and have verified of its performance on CIFAR-10 and SVHN akin to what is reported in (Luo et al., 2017). However, for CIFAR-100 dataset, since the experimental result when data augmentation is not used is not reported, we thus report our reproduced result.

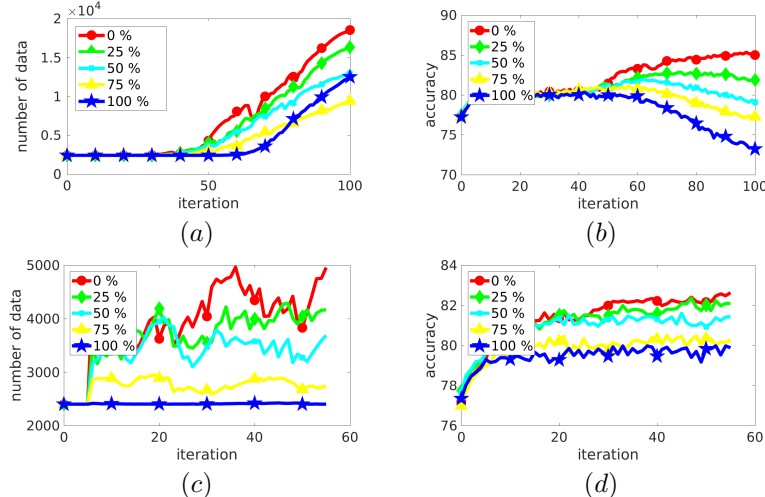

Figure 3: Result of new SSL problems on CIFAR-10 dataset with 5 runs. (a) number of data with iteration in decay mode (b) accuracy with iteration in decay mode (c) number of data with iteration in fixed mode(d) accuracy with iteration in fixed mode. % means the ratio of the number of non-animal classes in the unlabeled data.

Table 3: Classification error for new SSL problems on CIFAR-10 and CIFAR-100 dataset with 5 runs. '%' means the ratio of the number of non-animal classes.

| dataset | CIFAR-10 | | CIFAR-100 | |
|---|---|---|---|---|
| method | SST(decay) | SST(fixed) | SST(decay) | SST(fixed) |
| supervised | $22.27 \pm 0.47\%$ | | $34.62 \pm 1.14\%$ | |
| 0% | $14.99 \pm 0.54\%$ | $17.84 \pm 0.39\%$ | $28.01 \pm 0.44\%$ | $32.16 \pm 0.64\%$ |
| 25% | $17.93 \pm 0.33\%$ | $18.38 \pm 5247\%$ | $29.94 \pm 0.45\%$ | $32.28 \pm 0.58\%$ |
| 50% | $20.91 \pm 0.53\%$ | $19.04 \pm 0.63\%$ | $31.78 \pm 0.62\%$ | $32.6 \pm 0.67\%$ |
| 75% | $22.72 \pm 0.42\%$ | $20.07 \pm 0.98\%$ | $34.44 \pm 0.85\%$ | $32.32 \pm 0.52\%$ |
| 100% | $26.78 \pm 1.35\%$ | $20.24 \pm 0.15\%$ | $37.17 \pm 1.08\%$ | $32.62 \pm 0.63\%$ |

conjectured that the hyper-parameter in the current temporal ensembling and SNTG may not have been optimized.

## 4.2 NEW SSL PROBLEMS WITH OUT-OF-CLASS UNLABELED DATA

We have experimented with the following settings for real-world applications. The dataset is categorized into six animal and four non-animal classes as similarly done in (Odena et al., 2018). In CIFAR-10, 400 images per animal class are used as the labeled data (total 2,400 images for 6 animal classes) and a pool of 20,000 images with different mixtures of both animal and non-animal classes are experimented as an unlabeled dataset. In CIFAR-100, 5,000 labeled data (100 images per animal class) and a total of 20,000 unlabeled images of both classes with different mixed ratios are utilized. Unlike the experimental setting in (Odena et al., 2018), we have experimented according to the ratio (%) of the number of out-of-class data in the unlabeled dataset. More details are provided in Section 6.6 in the supplementary material.

As mentioned in Section 4, in the presence of out-of-class samples, a strict threshold is required. If all of the unlabeled data is assumed to be in-class, the decay mode may be a good choice. However, in many real-applications, out-of-class unlabeled data is also added to the training set in the decay mode and causes poor performance. In avoidance of such matter, we have experimented on a *fixed mode* of criterion threshold on adding the unlabeled data. Unlike the decay mode that decrements the threshold value, SST in the fixed mode sets a fixed threshold at a reasonably high value throughout the training. Our method in the fixed mode should be considered more suitable for real-applications but empirically shows lower performances in Figure 3 and Table 3 than when running in the decay

mode. The difference between the decay mode and the fixed mode are an unchangeable $\epsilon$ and the initial ensemble.

Setting a threshold value for the fixed mode is critical for a feasible comparison against the decay mode. Figure 3 shows the average of the results obtained when performing SST five times for each ratio in CIFAR-10. As shown in Figure 3(a), as the number of iteration increases, the threshold in the decay mode decreases and the number of additional unlabeled data increases. Obviously, while the different percentage of the non-animal data inclusion show different trends of training, in the cases of $0 \sim 75\%$ of non-animal data included in the unlabeled dataset, the additionally selected training data shows an initial increase at $30^{th} \sim 40^{th}$ iteration. On the other hand, when the unlabeled dataset is composed of only the out-of-class data, selective data addition of our method initiates at $55^{th} \sim 65^{th}$ training iteration. This tendency has been observed in previous researches on classification problems and we have set the threshold value fixed at a value between two initiating points of data addition as similarly done in the works of(Viola & Jones, 2001; Zhang & Viola, 2008). We have set the fixed threshold based on 47th iteration (between 40 and 55). For a more reliable selection score, we have not added any unlabeled data to the new training set and have trained our method with the labeled data only for 5 iterations.

As it can be seen in Table 3, in the case of SST in the decay mode, the performance has been improved when the unlabeled dataset consists only in-class animal data, but when the unlabeled pool is filled with only out-of-class data, the performance is degraded. For the case of SST with a fixed threshold value, samples are not added and the performance was not degraded at 100% non-animal ratio as shown in Figure 3(c). Furthermore, at 0% of out-of-class samples in the pool, there is a more improvement in the performance than at 100 % of out-of-class samples while still being inferior to the improvement than the decay mode. Because less but stable data samples are added by SST with a fixed threshold, the performance is improved for all the cases compared to that of supervised learning. Therefore, it is more suitable for real applications where the origin of data is usually unknown.

## 5 CONCLUSION

We proposed selective self-training (SST) for semi-supervised learning (SSL) problem. Unlike conventional methods, SST selectively samples unlabeled data and trains the model with a subset of the dataset. Using selection network, reliable samples can be added to the new training dataset. In this paper, we conduct two types of experiments. First, we experiment with the assumption that unlabeled data are in-class like conventional SSL problems. Then, we experiment how SST performs for out-of-class unlabeled data.

For the conventional SSL problems, we achieved competitive results on several datasets and our method could be combined with conventional algorithms to improve performance. The accuracy of SST is either saturated or not depending on the dataset. Nonetheless, SST has shown performance improvements as a number of data increases. In addition, the results of the combined experiments of SST and other algorithms show the possibility of performance improvement.

For the new SSL problems, SST did not show any performance degradation even if the model is learned from in-class data and out-of-class unlabeled data. Decreasing the threshold of the selection network in new SSL problem, performance degrades. However, the output of the selection network shows different trends according to in-class and out-of-class. By setting a threshold that does not add out-of-class data, SST has prevented the addition of out-of-class samples to the new training dataset. It means that it is possible to prevent the erroneous data from being added to the unlabeled dataset in a real environment.

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

# 6 SUPPLEMENTARY MATERIAL

## 6.1 THE BASIC SETTINGS OF OUR EXPERIMENTS

The basic settings of our experiments are as follows. Different from (Laine & Aila, 2016; Luo et al., 2017), we use stochastic gradient descent (SGD) with a weight decay of $0.0005$ as an optimizer. The momentum weight for the ensemble of selection scores is set to $\alpha = 0.5$. Also, we do not apply mean-only batch normalization layer (Salimans & Kingma, 2016) and Gaussian noise. We follow the same data augmentation scheme in (Laine & Aila, 2016) consisting of horizontal flips and random translations. However, ZCA whitening is not used. In the supervised learning phase, we train our model using batch size 100 for 300 epochs. After that, in the retraining phase, we train using the same batch size for 150 epochs with the new training dataset. The learning rate starts from $0.1$. In the supervised learning phase, it is divided by 10 at the 150-th and 225-th epoch. In the retraining phase, it is divided by 10 at the 75-th and 113-th epoch.

The number of training iteration and thresholding $\epsilon$ are very important parameters in our algorithm and have a considerable correlation with each other. In the first experiment, the iteration number remains fixed and the growth rate of $\epsilon$ is adjusted so that the validation accuracy saturates near the settled iteration number. While the validation accuracy is evaluated using the cross-validation, we set the number of training iteration to be 100 so that the model is trained enough until it saturates. $\epsilon$ is increased in log-scale and begins at a very small value ($10^{-5}$) where no data is added. The growth rate of $\epsilon$ is determined according to when the validation accuracy saturates. The stopping criterion is that the accuracy of the current iteration reaches the average accuracy of the previous 20 steps. If the stopping iteration is much less than 100 times, the $\epsilon$ growth rate should be reduced so that the data is added more slowly. If the stopping iteration significantly exceeds 100 iterations, the $\epsilon$ growth rate should be increased so that the data is added more easily. We allow 5 iterations as a deviation from 100 iterations and the growth rate of $\epsilon$ is left unchanged in this interval. As a result, the $\epsilon$ is gradually increased in log-scale by 10 times every 33 iterations in CIFAR-10 and SVHN. In the case of CIFAR-100, the $\epsilon$ is increased by 10 times in log-scale every 27 iterations. In the second experiment, we leave the $\epsilon$ fixed and simply train the model until the stopping criteria are satisfied. Other details are the same as those of the first experiment.

## 6.2 NETWORK STRUCTURE

We used two types of networks. The network for training the synthetic dataset is shown in Table 7 and consists of two hidden layers with 30 nodes. The network structure for CIFAR-10, SVHN, and CIFAR-100 consists of convolutions, and its structure is shown in Table 5. We used standard batch normalization (Ioffe & Szegedy, 2015) and Leaky ReLU (Maas et al., 2013) with 0.1.

Table 4: Network structure for synthetic datasets.

| Type | Filter Shape | Output Size |
|---|---|---|
| input | (x,y) point | 2 |
| fully connected | $2 \times 30$ | 30 |
| fully connected | $30 \times 30$ | 30 |
| classification outputs | $30 \times$ classes (softmax) | classes |
| selecting outputs | $30 \times 1$ (sigmoid) | 1 |

## 6.3 DATA DETAILS

We have experimented with CIFAR-10, SVHN, and CIFAR-100 datasets that consist of $32 \times 32$ pixel RGB images. CIFAR-10 and SVHN have 10 classes and CIFAR-100 has 100 classes. Overall, standard data normalization and augmentation scheme are used. For data augmentation, we used random horizontal flipping and random translation by up to 2 pixels. In the case of SVHN, random horizontal flipping is not used. To show that the SST algorithm is comparable to the conventional SSL algorithms, we experimented with the popular setting (Laine & Aila, 2016; Miyato et al., 2017; Tarvainen & Valpola, 2017). The validation set in the cross-validation to obtain the reduction rate of epsilon is extracted from the training set by 5000 images. After the epsilon is obtained, all the training datasets are used. The following is the standard labeled/unlabeled split.

**CIFAR-10 :** 4k labeled data ( 400 images per class ), 46k unlabeled data ( 4,600 images per class ),

Table 5: Network structure for CIFAR-10, SVHN, and CIFAR-100 datasets.

| Type | Filter Shape / pad | Output Size |
|---|---|---|
| input | RGB channel images | $32 \times 32 \times 3$ |
| conv2d | $3 \times 3 \times 3 \times 128$ / same | $32 \times 32 \times 128$ |
| conv2d | $3 \times 3 \times 128 \times 128$ / same | $32 \times 32 \times 128$ |
| conv2d | $3 \times 3 \times 128 \times 128$ / same | $32 \times 32 \times 128$ |
| pool | maxpool $2 \times 2$ | $16 \times 16 \times 128$ |
| dropout | p = 0.5 | $16 \times 16 \times 128$ |
| conv2d | $3 \times 3 \times 128 \times 256$ / same | $16 \times 16 \times 256$ |
| conv2d | $3 \times 3 \times 256 \times 256$ / same | $16 \times 16 \times 256$ |
| conv2d | $3 \times 3 \times 256 \times 256$ / same | $16 \times 16 \times 256$ |
| pool | maxpool $2 \times 2$ | $8 \times 8 \times 256$ |
| dropout | p = 0.5 | $8 \times 8 \times 256$ |
| conv2d | $3 \times 3 \times 256 \times 512$ / valid | $6 \times 6 \times 512$ |
| conv2d | $3 \times 3 \times 512 \times 256$ / same | $6 \times 6 \times 256$ |
| conv2d | $3 \times 3 \times 256 \times 128$ / same | $6 \times 6 \times 128$ |
| pool | average pool $6 \times 6$ | $1 \times 1 \times 128$ |
| classification outputs | $128 \times$ classes (softmax) | $1 \times 1 \times$ classes |
| selecting outputs | $128 \times 1$ (sigmoid) | $1 \times 1 \times 1$ |

and 10k test data.

**SVHN :** 1k labeled data ( 100 images per class), 72,257 unlabeled data (it is not well balanced), and 26,032 test data.

**CIFAR-100 :** 10k labeled data (100 images per class ), 40k unlabeled data ( 400 images per class ), and 10k test data.

## 6.4 SYNTHETIC DATASETS

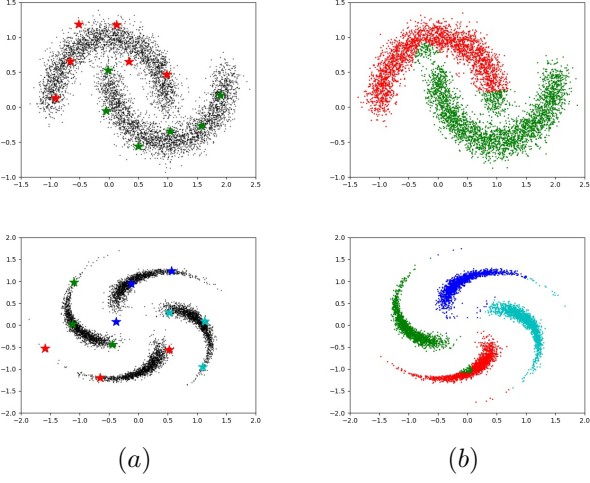

$(a)$ $(b)$

Figure 4: Synthetic datasets (Top : Two moons, Bottom : 4 spins) (a): training dataset (black point : unlabeled data point, color point : labeled data point), (b) test dataset result.

As synthetic datasets, two moons and 4 spins were tested in the same manner as SNTG(Luo et al., 2017). Each dataset has 6,000 training and 6,000 test samples. In the case of two moons, there are two classes $y \in \{0, 1\}$, and in case of 4 spins, $y \in \{0, 1, 2, 3\}$. In 6,000 training data, there are 12 labeled data and 5,988 unlabeled data. Thus, for two moons, each class has 6 points and for 4 spins, each class has 3 points. Because the number of labeled datapoints are too small, random sampling can lead to sample similar points. Therefore, we randomly sampled the labeled data with a constraint that the Euclidian distance of each data point is greater than 0.7. For these datasets, total iteration was performed 50 times, and the $\epsilon$ was increased from $10^{-7}$ to $10^{-4.5}$ on a log scale.

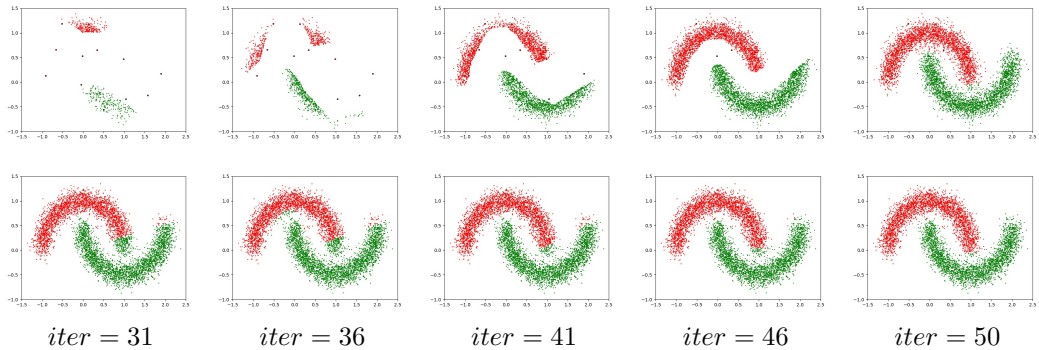

Figure 5: Synthetic datasets (Two moons) Top : training dataset, Bottom : test result

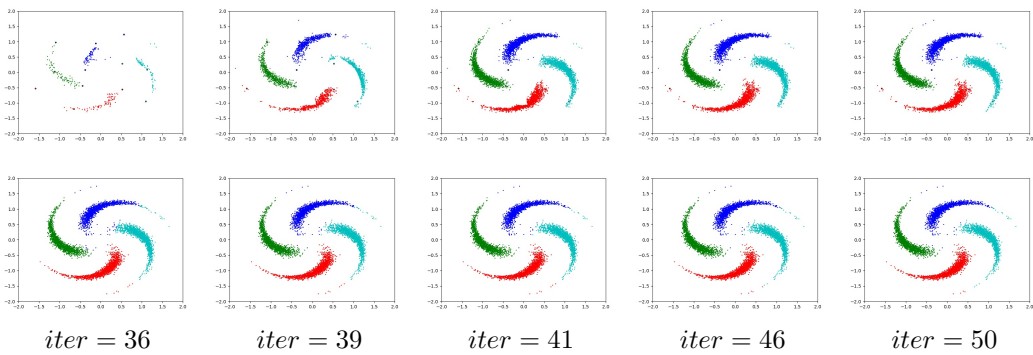

Figure 6: Synthetic datasets (Four spin) Top : training dataset, Bottom : test result

Figure 4 shows the basic setting of the synthetic dataset, and Figure 5 and 6 show the progress of the SST algorithm. The SST algorithm improves performance by gradually expanding certain data in a synthetic dataset.

## 6.5 FURTHER EXPERIMENTS

**CIFAR-10 :** When the network were trained with 1k and 2k images, the test error were 38.71% and 26.99% respectively. The test errors in the SST algorithm were 23.15% and 15.72%, the SST has better performance than Π model but worse than Mean Teacher in 1k test. In 2k test, the SST has better performance than Π model and similar with Mean Teacher.

Table 6: Classification error on CIFAR-10 (1k and 2k Labels) with 5 runs using in-class unlabeled data

| Method | CIFAR-10 (1k) | CIFAR-10 (2k) |
|---|---|---|
| supervised (sampled) | $38.71 \pm 0.47\%$ | $26.99 \pm 0.79\%$ |
| Π model (Laine & Aila, 2016) | $27.36 \pm 1.20\%$ | $18.02 \pm 0.60\%$ |
| Mean Teacher (Tarvainen & Valpola, 2017) | $21.55 \pm 1.48\%$ | $15,73 \pm 0.31\%$ |
| Proposed method (SST) | $23.15 \pm 0.61\%$ | $15.72 \pm 0.50\%$ |

**SVHN :** For the balancing experiments, in SVHN, 1,000 images are used as the labeled data and 45,000 balanced unlabeled images are used. As a result, the SST is still worse than other algorithms. As mentioned in Section 4.1, we think that incorrectly estimated samples by SST deteriorate the performance.

Table 7: Classification error on SVHN (balanced & all unlabeled data) with 5 runs using in-class unlabeled data.

| Method | SVHN (balanced) | SVHN (all) |
|---|---|---|
| supervised (sampled) | $13.45 \pm 0.92\%$ | |
| Π model (Laine & Aila, 2016) | $5.09 \pm 0.31\%$ | $4.82 \pm 0.17\%$ |
| TempEns (Laine & Aila, 2016) | $5.01 \pm 0.15\%$ | $4.42 \pm 0.16\%$ |
| Proposed method (SST) | $6.75 \pm 0.28\%$ | $6.88 \pm 0.59\%$ |

## 6.6 Data setting for New SSL problem

(Odena et al., 2018) adds only four unlabeled classes and tests according to the radio of unlabeled class. For example, at 50%, two classes are in-class, and two classes are out-of-class. However, we experimented with the ratio of the number of non-animal data. Thus at 50% in CIFAR-10, unlabeled data consists of 50% in-class and 50% out-of-class. The data for each ratio are shown in Table 8, and the data category for animal and non-animal is shown in Table 9.

Table 8: Number of each class data for new SSL problems.

| Dataset | Ratio | Labeled data | Unlabeled data | |
|---|---|---|---|---|
| | | Animal | Animal | Non-Animal |
| CIFAR-10 | 0% | 400 | 3334 or 3333 | 0 |
| | 25% | 400 | 2500 | 1250 |
| | 50% | 400 | 1667 or 1666 | 2500 |
| | 75% | 400 | 834 or 833 | 3750 |
| | 100% | 400 | 0 | 5000 |
| CIFAR-100 | 0% | 100 | 400 | 0 |
| | 25% | 100 | 300 | 100 |
| | 50% | 100 | 200 | 200 |
| | 75% | 100 | 100 | 300 |
| | 100% | 100 | 0 | 400 |

Table 9: Data category for new SSL problems.

| Type | Animal | Non-Animal |
|---|---|---|
| CIFAR-10 (CLASS) | bird, cat, deer, dog, frog, horse | airplane, automobile, ship, truck |
| CIFAR-100 (SUPERCLASS) | aquatic mammals, fish, insects, large carnivores, large omnivores and herbivores, medium-sized mammals, non-insect invertebrates, people, reptiles, small mammals | flowers, food containers, fruit and vegetables, household electrical devices, household furniture, large man-made outdoor things, large natural outdoor scenes, trees, vehicles 1, vehicles 2 |

## 6.7 Other algorithms for new SSL problems

Table 10 shows the results of a general test on other algorithms. First, self-training (McLachlan, 1975; Zhu, 2007; Zhu & Goldberg, 2009) without threshold does not improve performance even at 0%, and performance at 100% is degraded. When SST is applied to the softmax output as a threshold without selection network, the performance is improved at 0%, but the performance is degraded at 100%. Although the threshold was 0.9999, unlabeled data was added in 100% of the non-animal data.

## 6.8 New SSL problems in CIFAR-100

In the new SSL problem, the experiment in decay mode is to find a gap between two initiating points of data addition in $0 \sim 75\%$ of non-animal data and 100% of non-animal data. In our experiment, the growth rate of epsilon in CIFAR-10 is applied to CIFAR-100. (The smaller the growth rate of epsilon, the less the difference in $\epsilon$ between iterations. Therefore, although the difference in the $\epsilon$

Table 10: Other algorithms for new SSL problems of increased classes with 5 runs.

| method | self-training | SST (softmax) | SST (sigmoid) |
|---|---|---|---|
| supervised | | $22.27 \pm 0.47\%$ | |
| 0% | $21.97 \pm 0.24\%$ | $18.27 \pm 0.52\%$ | $17.84 \pm 0.39\%$ |
| 25% | $22.80 \pm 0.39\%$ | $18.35 \pm 0.86\%$ | $18.38 \pm 0.52\%$ |
| 50% | $23.93 \pm 0.71\%$ | $18.72 \pm 0.36\%$ | $19.04 \pm 0.63\%$ |
| 75% | $25.45 \pm 0.47\%$ | $20.33 \pm 0.82\%$ | $20.07 \pm 0.98\%$ |
| 100% | $27.31 \pm 0.57\%$ | $20.71 \pm 0.19\%$ | $20.24 \pm 0.15\%$ |

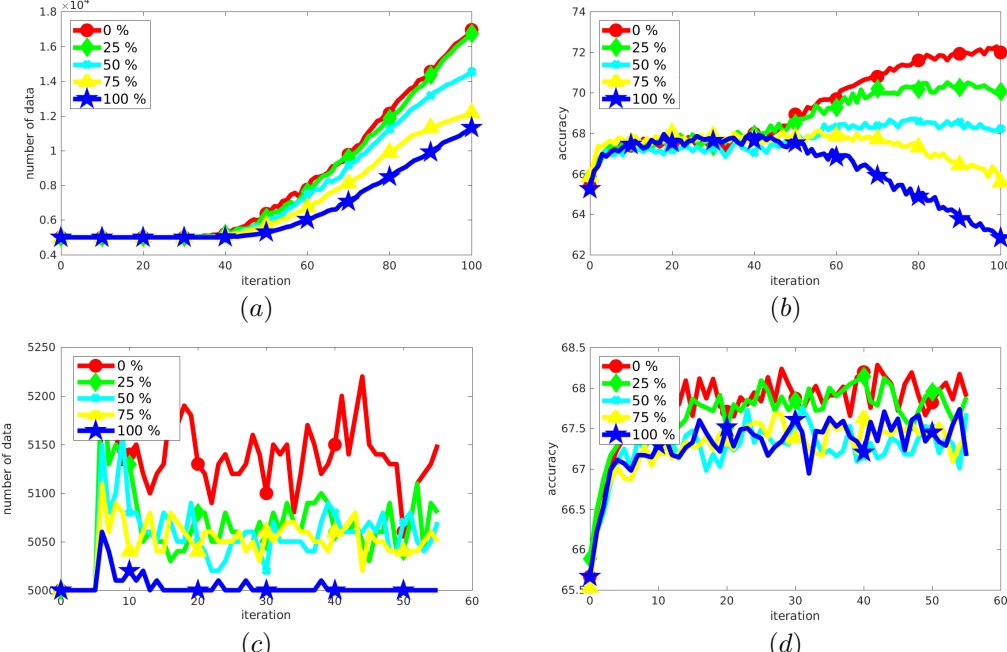

Figure 7: Result of new SSL problems on CIFAR-100 dataset with 5 runs. (a) number of data with iteration in decaying threshold (b) accuracy with iteration in decaying threshold (c) number of data with iteration in fixed threshold (d) accuracy with iteration in fixed threshold. % means the ratio of non-animal classes in the unlabeled data.

between intervals is the same, depending on the growth rate of $\epsilon$, the difference in the iteration can be greater.) In the case of $0 \sim 75\%$, the number of data shows a slight increase from about 30 iterations. On the other hand, in the case of 100%, selected samples are added from about 40 iterations. The fixed threshold set to the threshold of 35 iterations. In the decay mode, the performance is much improved at 0%, and at 100%, the performance is degraded. On the other hand, in the fixed mode, there was no performance degradation from 0% to 100%. In CIFAR-100, the difference between 0% and 100 % is less than CIFAR-10, because the gap between animal and non-animal is small and additional data is small. Figure 7 shows the experimental results.

