# OpenReview forum: "Selective Self-Training for semi-supervised Learning"
_ICLR.cc/2019/Conference_

### Official Review · AnonReviewer3 · 2018-11-02
**Review: how principled the method is and how experimental evaluation confirms the benefits and applicability of the method**

**Rating:** 4
**Confidence:** 4

**Review:**

This paper describes the method for performing self-training where the unlabeled datapoints are iteratively added to the training set only if their predictions by the classifier are confident enough. The contributions of this paper are to add datapoints based on the prediction of the confidence level by a separate selection network and a number of heuristics applied for better selection. On the experimental side, the contribution is to test the scenario where datapoints from irrelevant classes are included in the unlabeled dataset.
The paper is written in a way that makes following it a bit difficult, for example, the experimental setups. Also, the writing can be improved by making the writing more concise and formal (examples of informal: "spoil the network", "model is spoiled", "problem of increased classes", "many recent researches have been conducted", "lots of things to consider for training", "supervised learning was trained" etc.). The contributions of the method could also be underlined more clearly in the abstract and introduction. The description of consistency regularisation methods in section 2.2 is not very clear and I would like to get better understanding of temporal ensembling and SNTG methods here as they play an important role in the experiments.
The idea of selective sampling for self-training is promising and the investigated questions are interesting. As far as I understand, the main contribution of this paper is the use of separate "selection network" to estimate the confidence of predictions by "classification network". However, as the "selection network" uses exactly the same input as "classification network", it is hard to imagine how it can learn additional information. For example, imagine the case of binary classification. If the selection network predicts 0 in come cases, it can be used to improve the result of "classification network" by flipping the corresponding label. How can you interpret such a thought experiment? One could understand the use of "selection network" as a way to automatically select a threshold of what to consider confident, however, in this case, the prediction of "selection network" should be thresholded at 0.5 (correct prediction or not), but the experiments use complex thresholds. Could you elaborate more on why the selection network is needed? How would it compare to a simple strategy of only including the datapoints whose top-1 prediction of "classification network" is greater than some threshold? Finally, could you show a plot of top-1 prediction of "classification network" vs score of "selection network" and elaborate on that?
Then, in sections 3.2 and 3.3 the authors introduce a few additional tricks for self-training: exclude datapoints whose predictions are changing and balance the classes. Intuitively, these criteria are well motivated, but unfortunately, the combination of all the intuitions (including "selection network" with threshold) is not very principled. Ablation study shows that the use of the "selection network" strategy does not improve the results without these heuristics. It would be interesting to see how these heuristics would do without "selection network", for example, either by doing simple self-training with thresholding on the score of the classifier or by applying only these heuristics in combination with TempEns+SNTG. In the current form of evaluation, it is hard to say if there is any benefit of using the "selection network" that is the main novelty of the paper.
It is very valuable that the experimental results include many recently proposed methods. Besides, the settings are described in details that could help for the reproducibility of the results. However, I have a few concerns about the results. First of all, the proposed SST algorithm alone only performs better than baselines in 1 case, equal to them in 1 case and worse in 1 (table 3). Besides, as the base classifier is different for various baselines, it is hard to compare the methods. Then, the important hyperparameter of the method---threshold---seems to be hard to select (both in sections 4.1 and 4.2). How did you chose the current values? How sensitive is it? Why various datasets need different settings? How the threshold value can be set in practice? Another important parameters is the number of iterations of the algorithm. How was it chosen? Concerning the experiments of section 4.2, how would the baseline methods of section 4.1 do in this case? Why did you select to study animal vs non-animals sets of classes? What would happen if you use random class splits or split animal classes (like in a more realistic scenario)?
To conclude, while I find the studied problem quite interesting and intuitions behind the method very reasonable, the current methodology is not very principled and the experiment evaluation did not convince me that such an elaborate strategy is needed.

Some questions and comments:
- The setting of including unrelated classes in the unlabeled data resembles transfer learning setting. Could you explain why the ideas from transfer learning are not applicable in your case?
- In the training procedure of "selection network" of Sections 3.1, do you use the same datapoints to train a "classification network" and "selection network"? If it is the case, how do you insure that the "classification network" does not learn to fit the data perfectly and thus all labels s_i are 1?
- In the last sentences of the first paragraph on p.2 you make a contrast between using softmax and sigmoid functions, however, normally the difference between them is their use in binary or multiclass classification. Is there anything special that you want to show in you case?
- What do you mean in section 3.3 by "if one class dominates the dataset, the model tends to overfit"?
- I think parameters of training the networks from the beginning of section 4 could be moved to the supplementary materials.
- Figure 3: wouldn't the plot of accuracy vs amount of data be more suitable here?
- Synthetic experiments of supplementary materials: the gains of the methods seem to be small. What are the numerical results? What would happen if you allow to select starting point at random (a more realistic case)?
- Can you explain the sentence "To prevent data being added suddenly, no data was added until 5 iterations"?
- How was it possible to improve the performance in experiment of section 4.2 with 100% of irrelevant classes?

---

> ### Author Response · Authors · 2018-11-26
> **Response to reviewer 3 (part 1)**
>
> First of all, thank you for taking your time to review our paper and providing feedback. We have judiciously taken the comments of the reviewers,  and apologize for the late response due to additional experiments and modifications of the paper.
>
>
> Remark 1. Expression and detail
>
> A : We apologize to the reviewer for the lack of clarity in the manuscript. We have modified our expression.
>
>
> Remark 2. What is "Selection Network"?
>
> A : It is a module that estimates the confidence of the softmax output according to the inputs of the classification network. The selection network is trained with sigmoid and binary cross-entropy in a supervised manner. And the threshold is not 0.5 but high because selection network is learned with many ’1’ labels with close to 100 % training accuracy.
> The selection network has advantages in out of class unlabeled data. Since softmax output is a relative value, the softmax output can be high for some out of class unlabeled data. In our original paper (in table 10), there already exist results of softmax output for in or out of class unlabeled data with 0.9999 thresholds. Further, we experimented with the same threshold in table 4 of the new version and the results have shown that out of class unlabeled data are added even with an extremely small threshold such as 0.99999 (epsilon = 10^-5).
>
>
> Remark 3. "As the base classifier is different for various baselines, it is hard to compare the methods."
>
> A : SST has a network structure similar to other papers. The difference of structure was that the selection network is added and Gaussian noise and the mean only batch norm are not used. As mentioned in the paper (4. Experiments), our supervised learning performs slightly better than conventional SSL algorithms because of different settings such as learning rate and Gaussian noise on the input layer. (When SST uses Gaussian noise, ours are also degraded.)
>
>
> Remark 4. Experiments Detail ( data setting, threshold, number of iterations, animal vs nonanimal)
>
> A :
> ==> Data setting
> The purpose of experiments is to show that the SST algorithm is comparable to the conventional SSL algorithms. Therefore, we experimented with the popular setting. We have added a detailed description on the data setting to Section 6.3 of the supplementary material.
>
>
> ==> Iterations & Threshold
> We have missed out on a detailed description of how to set up some hyper-parameters. We set parameters as follows. The number of training iteration and thresholding epsilon are very important parameters in our algorithm and have a considerable correlation with each other.
>
> In the first experiment, the iteration number remains fixed and the growth rate of epsilon is adjusted so that the validation accuracy saturates near the settled iteration number. While the validation accuracy is evaluated using the cross-validation, we set the number of training iteration to be 100 so that the model is trained enough until it saturates. Epsilon is increased in log-scale and begins at a very small value (10^(−5)) where no data is added. The growth rate of epsilon is determined according to when the validation accuracy saturates. The stopping criterion is that the accuracy of the current iteration reaches the average accuracy of the previous 20 steps. If the stopping iteration is much less than 100 times, the epsilon growth rate should be reduced so that the data is added more slowly. If the stopping iteration significantly exceeds 100 iterations, the epsilon growth rate should be increased so that the data is added more easily. We allow 5 iterations as a deviation from 100 iterations and the growth rate of epsilon is left unchanged in this interval. (In previous versions, the growth ratio of epsilon for CIFAR-10 was applied to SVHN and CIFAR-100. However, since the epsilon growth rate is different for each dataset, as the reviewer mentioned, we have performed the cross-validation for SVHN and CIFAR-100 and modified our results.) As a result, the epsilon is gradually increased in log-scale by 10 times every 33 iterations in CIFAR-10 and SVHN. In the case of CIFAR-100, the epsilon is increased by 10 times in log-scale every 27 iterations. In the second experiment, we leave the epsilon fixed and simply train the model until the stopping criterion is satisfied. Other details are the same as those of the first experiment. (In previous versions, the training iterations of fixed mode had been fixed. Thanks to the comment from the reviewer, we were able to rearrange the content and set training iteration by cross-validation.)
>
>
> ==> Animal vs non-animal
> The citation of that part is obscure and has been modified. We experimented similar to the [1] and they categorized according to the animal. Our approach is similar but not identical. Their unlabeled data came from only in 4 classes, however, we selected unlabeled data in all classes.
> [1] Odena, Augustus, et al. "Realistic Evaluation of Semi-Supervised Learning Algorithms." (2018)

---

> > ### Author Response · Authors · 2018-11-26
> > **Response to reviewer 3 (part 2)**
> >
> > Some Questions and comments
> >
> > Remark 5. "The setting of including unrelated classes in the unlabeled data resembles transfer learning setting. Could you explain why the ideas from transfer learning are not applicable in your case?"
> >
> > A : To the best of our knowledge, the main purpose of transfer learning is to improve the performance on the target domain by effectively utilizing the knowledge of the source domain. However, in our case, there is no separated source and target domains. We focus on the single classification task. We think that the goal of our method and that of transfer learning are quite different.
> >
> >
> > Remark 6. "What do you mean in section 3.3 by "if one class dominates the dataset, the model tends to overfit"?"
> >
> > A " We have modified that expression and we wanted to address that "if one class dominates the dataset, the performances are degraded by the imbalanced distribution. (Analysing the classification of imbalanced data-sets with multiple classes: Binarization techniques and ad-hoc approaches, 2013)"
> >
> >
> > Remark 7. "Figure 3: wouldn’t the plot of accuracy vs amount of data be more suitable here?"
> >
> > A : I agree that your suggestion is more suitable for the figure. However, it is difficult to show the figure you want because the number of selected samples is different every time.
> >
> >
> > Remark 8. "Synthetic experiments of supplementary materials: the gains of the methods seem to be small. What are the numerical results? What would happen if you allow to select starting point at random (a more realistic case)?"
> >
> > A : The performance depends on the initial points, therefore sometimes the performance is not good. Since the inputs are the x and y coordinate values, it can be very easy to add to the training set. (ex.. class 1 : (-1, 0), (1, 0) , class 2 : (0.5, -0.5), (1.5, -0.5) , then decision boundary could be (:, -0.25) then class 2 unlabeled data (0, 0.5) is classified as class 1 and can have a very high selection score.)
> >
> >
> > Remark 9. Can you explain the sentence "To prevent data being added suddenly, no data was added until 5 iterations"?
> >
> > A : In fixed mode, we ensemble the selection scores, which makes the prediction more consistent. Also, for a more reliable selection score, we do not add unlabeled data to the new training set and train with labeled data only for 5 iterations.
> >
> >
> > Remark 10. "How was it possible to improve the performance in the experiment of section 4.2 with 100% of irrelevant classes?"
> >
> > A : We suspect that this performance improvement is due to re-initializing learning rate. After constructing a new training dataset, we retrain our model with the learning rate of the initial value. In decay mode (Figure 2, Figure 3 (a) and (b) of the original manuscript), the accuracy is slightly increased and gets saturated while unlabeled data is not being added. However, the accuracy begins to increase or decrease relatively more after adding selected data to the new training dataset. In fixed mode (Figure 3 (c) and (d) of the original manuscript), the improvement with the 100% of irrelevant classes seems to be due to re-initializing learning rate. However, SST algorithm with other ratios of out-of-class samples results in performance improvement compared to the 100% because out-of-class samples are not selected.

---

### Official Review · AnonReviewer1 · 2018-11-03
**Selective Self-Training for semi-supervised Learning**

**Rating:** 5
**Confidence:** 4

**Review:**

This is an novel, interesting paper on an important topic: semi-supervised learning.
Even though the proposed approach seems to have significant potential, the experimental
is somewhat disorganized,  and it also includes some weak claims that should be removed.

For example, the number of labeled examples in Table 1 is fairly large and inconsistent (4K, 1K, 10K for the 3 organic datasets).  In this reviewer's opinion, it would be a lot more reasonable to have instead a learning curve showing the results for, say, 100, 500, 1K, 5K, and 10K labeled examples for all three domains.

In 4.1, you are using different epsilon policies for synthetic vs organic datasets; why?

The explanation for underperforming on SVHM (page 7) may be valid, but you could easily prove it right or wrong by adding an option to SST for "stratified SSL." Without this extra work, your claim is just a conjecture.

You should also show the performance of regular SSL methods in the setup on Table 4.

Last but not least, you have repeatedly made the claim combining SST and other SSL may further improve the performance;
however, you do not provide any evidence for it, so you should avoid making such claims.

Other comments:
- on page 2, the two terms classification & selection network appear "out of the blue;" it would be quite helpful to make it clear from the abstract that the proposed implementation is for neural networks.
- figures 2 & 3 should be a lot larger in order to be readable
- 4.1.2 top of page 7: claims such as "SST could have obtained better performance" have no place in such a paper; you could instead make a note about the method being "prohibitively CPU intensive for the time being"
- lower on the same page you say: "SST may get better performance" - see above

---

> ### Author Response · Authors · 2018-11-26
> **Response to reviewer 2**
>
> First of all, thank you for taking your time to review our paper and providing feedback. We have judiciously taken the comments of the reviewers,  and apologize for the late response due to additional experiments and modifications of the paper.
>
>
> Remark 1. "the number of labeled examples in Table 1 is fairly large and inconsistent (4K, 1K, 10K for the 3 organic datasets)."
>
> A : The purpose of our experiments is to show that the SST algorithm is comparable to the conventional SSL algorithms. Therefore, we experimented with the popular setting. The following is the experimented dataset in other papers.
>
> - Temporal ensembling & Π model [1]: CIFAR-10 (4k), SVHN (500, 1k), CIFAR-100 (10k)
> - VAT [2] : CIFAR-10 (4k), SVHN (1k), CIFAR-100 (no experiment)
> - Mean Teacher [3]: CIFAR-10 (1k, 2k, 4k), SVHN (250, 500, 1k), CIFAR-100 (10k)
>
> We took the reviewer’s comment judiciously and have added CIFAR-10 (1k, 2k) experiments in Section 6.5 of the supplementary material and their accuracies are comparable with those of the conventional SSL algorithms. (It took a long time to perform 5 runs of test for all additional experiments.)
>
> ———————————————————————————
> The result of CIFAR-10 (1k, 2k) with 5 runs
> ———————————————————————————
> (error     / standard deviation) |        1k        |        2k
> supervised                       | ( 38.71 / 0.47 ) | ( 26.99 / 0.79 )
> Π model [1]                      | ( 27.36 / 1.2 )  | ( 18.02 / 0.60 )
> mean teacher [3]                 | ( 21.55 / 1.48 ) | ( 15.73 / 0.31 )
> SST                              | ( 23.15 / 0.61 ) | ( 15.72 / 0.50 )
> ———————————————————————————
>
> [1] Laine, Samuli, and Timo Aila. "Temporal ensembling for semi-supervised learning." arXiv preprint arXiv:1610.02242 (2016).
>
> [2]Takeru Miyato, Shin-ichi Maeda, Masanori Koyama, and Shin Ishii. Virtual adversarial training: a regularization method for supervised and semi-supervised learning. arXiv preprint arXiv:1704.03976, 2017.
>
> [3] Antti Tarvainen and Harri Valpola. Mean teachers are better role models: Weight-averaged consistency targets improve semi-supervised deep
>
>
> Remark 2. "Why does the SST algorithm use different epsilon policies for synthetic vs organic datasets?"
>
>
> A : There are different network structures in the synthetic and organic dataset (table 4-5 in the supplementary materials in the new version). And, because there are only 12 initial points in the synthetic, it needs much higher confidence than organic datasets. (Epsilon is increased in log-scale and begins at a very small value (10^(−5)) where no data is added in the organic dataset. However, in synthetic data, unlabeled data is added when epsilon begins at (10^(−5)). Therefore, We have changed the epsilon value so that no data is added at the beginning of the iteration.)
>
>
> Remark 3. What is the problem in SVHN (balance problem or dataset or both)?
>
> A : We have experimented with SVHN with data balancing. In SVHN, 1,000 images are used as the labeled data and 45,000 balanced unlabeled images are used. As a result, the SST is still worse than other algorithm [1]. Therefore, we have modified our expression and added the result in Section 6.4 of the supplementary material.
>
>
> Remark 4. "You should also show the performance of regular SSL methods in the setup on Table 4."
>
> A : We have performed experiments of self-training without threshold and SST with softmax output. Although the experimental setting is a bit different from [1], the setting of 100% of non-animal unlabeled data is the same. They have shown that the performance degraded when the unlabeled dataset contained 100% of non-animal data in figure 2 in [4].
>
> The approximate score in Figure 2 of [4]
> —————————————————————
> 100% out-of-class (error)
> —————————————————————
> supervised learning : about 23.5%
> Π model : about 26.3 %
> Mean Teacher : about 26.3 %
> VAT : about 26%
> —————————————————————
> [4] Odena, Augustus, et al. "Realistic Evaluation of Semi-Supervised Learning Algorithms." (2018). ( https://arxiv.org/abs/1804.09170 )
>
>
> Remark 5. combining SST and other SSL
>
> A : Combining and the additional cost is expensive. Therefore, we have modified our expressions.

---

### Official Review · AnonReviewer2 · 2018-11-08
**Review of SST for SSL**

**Rating:** 4
**Confidence:** 5

**Review:**

Summary:
In the semi-supervised self-training setting, this paper proposes to select a certain subset of unlabelled data for training rather than all unlabelled data, where the ensemble of confidence scores of the trained model in iterations is used to guide the selection.

Strong points:
It is a good idea to conduct an ensemble based on the confidence scores of trained models in iterations, although the authors did not mention any theoretical explanation or guarantee behind this.

Weak points:
1) Although the ensemble idea is new, the idea of selective self-training is not novel in self-training or co-training of SSL as in the following survey. Considering the selection based on highest-confidence, the in or out of class unlabeled data in most cases does not matter. Therefore, the technical contribution of this paper is moderate.

Zhu, Xiaojin. "Semi-supervised learning literature survey." Computer Science, University of Wisconsin-Madison 2.3 (2006): 4.

 2) The writing is poor and hard to follow. First, many details are missing, especially in the experiments, which makes the proposed method suspicious and non-convincing. For example, what is the number_iterations in the experiments? How are they chosen or what's the specific stopping criteria? From the plots in Figure 2 and 3, it is hard to find the convergence of the method within 100 iterations. The descriptions of the datasets used are not clear, e.g., the number of classes for each data. Second, many typos and grammar errors need to fix, e.g., "the proposed SST is suitable for lifelong learning which make use...", "the error 21.44% was lower than" 18.97?

3) The overall performance of the proposed SST in the experiments is not convincing and not promising. First, the labeled data portion is fixed and is relatively high compared to most standard semi-supervised learning settings. Second, SST itself is only comparable with or even worse than the state-of-art methods. Combining SST with other existing techniques can help. However, the additional cost is expensive. Further demonstrations are necessary for the proposed SST method.

---

> ### Author Response · Authors · 2018-11-26
> **Response to reviewer 1 (part 1)**
>
> First of all, thank you for taking your time to review our paper and providing feedback. We have judiciously taken the comments of the reviewers, and apologize for the late response due to additional experiments and modifications of the paper.
>
>
> Remark 0. It needs other theoretical explanation (ex. co-training)
>
> A: We have modified our paper and added some theoretical explanation in the introduction on page 2.
>
>
> Remark 1. " Considering the selection based on highest-confidence, the in or out of class unlabeled data in most cases does not matter. "
>
> A: We do not agree with your opinion. The formulae of the softmax and sigmoid are as follows.
>
> The softmax function : exp(f_j(z)) / sigma(exp(f_k(z))
> = 1 / ( 1 + exp(-f_j(z)) × (exp(f_1(z)) + ... + exp(f_j-1(z)) + exp(f_j+1(z)) + ... exp(f_n(z)))
>
> The sigmoid function : 1 / (1 + exp(-g(z)))
>
> where z, f(z), and g(z) represent the final layer of the backbone network, classification network, and selection network respectively. As you said, if f_j(z) is very high and the other f(z)s are moderate, it can work like sigmoid. However, even if f_j(z) is not much high, the softmax output can be close to 1 with extremely smaller values for other f(z)s because:
>
> The softmax output : 1 / ( 1 + exp(-f_j(z)) × 0) = 1
>
> We experimented with a high softmax output threshold (epsilon = 10^(−4)). Although epsilon was 10^(−4) (threshold = 0.9999), an average of about 800 unlabeled data was added for the case of 100% of the non-animal data. Even at 0% of the non-animal data, performance is lower than the fixed mode of the sigmoid. This shows the limitation of thresholding with softmax.
>
> The result of new SSL problems on the CIFAR-10 dataset with 5 runs are as follows:
> ————————————————————————————
> activation function / softmax (error/added data) / sigmoid (error/added data)
> ————————————————————————————
> supervised    /    ( 22.27 / 0 )
> ————————————————————————————
> 0%                  /    (18.27 / 4,306.8)       /   (17.84/2,338.8 )
> 25%                 /    (18.35 / 3,350.4 )      /   (18.38 / 1470.0 )
> 50%                 /    ( 18.72 / 2580.0 )      /   (19.04 / 811.2 )
> 75%                 /    (20.33 / 1,711.2 )      /   ( 20.07 / 315.6 )
> 100%                /    (20.71 / 864.0 )        /   ( 20.24 / 1.2 )
> ————————————————————————————
>
>
> Remark 2. Expression, and details (ex. number of iterations, stopping criteria, typos and grammar errors)
>
> A : We apologize to the reviewer for the lack of clarity in the manuscript. We have modified our expression, typos and grammar errors.
>
> Regarding the details on hyper-parameters:
> - We set parameters as follows. The number of training iteration and thresholding epsilon are very important parameters in our algorithm and have a considerable correlation with each other.
>
> In the first experiment, the iteration number remains fixed and the growth rate of epsilon is adjusted so that the validation accuracy saturates near the settled iteration number. While the validation accuracy is evaluated using the cross-validation, we set the number of training iteration to be 100 so that the model is trained enough until it saturates. Epsilon is increased in log-scale and begins at a very small value (10^(−5)) where no data is added. The growth rate of epsilon is determined according to when the validation accuracy saturates. The stopping criterion is that the accuracy of the current iteration reaches the average accuracy of the previous 20 steps. If the stopping iteration is much less than 100 times, the epsilon growth rate should be reduced so that the data is added more slowly. If the stopping iteration significantly exceeds 100 iterations, the epsilon growth rate should be increased so that the data is added more easily. We allow 5 iterations as a deviation from 100 iterations and the growth rate of epsilon is left unchanged in this interval. (In previous versions, the growth ratio of epsilon for CIFAR-10 was applied to SVHN and CIFAR-100. However, since the epsilon growth rate is different for each dataset, as the reviewer mentioned, we have performed the cross-validation for SVHN and CIFAR-100 and modified our results.) As a result, the epsilon is gradually increased in log-scale by 10 times every 33 iterations in CIFAR-10 and SVHN. In the case of CIFAR-100, the epsilon is increased by 10 times in log-scale every 27 iterations.
>
> In the second experiment, we leave the epsilon fixed and simply train the model until the stopping criterion is satisfied. Other details are the same as those of the first experiment. (In previous versions, the training iterations of fixed mode had been fixed. Thanks to the comment from the reviewer, we were able to rearrange the content and set training iteration by cross-validation.)

---

> > ### Author Response · Authors · 2018-11-26
> > **Response to reviewer 1 (part 2)**
> >
> > Remark 3. "SST itself is only comparable with or even worse than the state-of-art methods."
> >
> > A : As mentioned in our paper, SST has comparable performance to other conventional SSL algorithms. In Table 2 of our paper, SST achieves 34.89% on CIFAR-100, which is higher than TempEns[1](38.65%), 11.82% on CIFAR-10, which is slightly worse than VAT+EntMin[2](10.55%), and perform worse 6.88% on SVHN. However, SST can solve the real problem of the existence of out-of-class unlabeled data.
> >
> > [1] Laine, Samuli, and Timo Aila. "Temporal ensembling for semi-supervised learning." arXiv preprint arXiv:1610.02242 (2016).
> >
> > [2] Miyato, Takeru, et al. "Virtual Adversarial Training: a Regularization Method for Supervised and Semi-supervised Learning." arXiv preprint arXiv:1704.03976 (2017).
> >
> >
> > Remark 4. "Combining SST with other existing techniques can help. However, the additional cost is expensive. Further demonstrations are necessary for the proposed SST method."
> >
> > A : It is true that combining and the additional cost is expensive. Therefore, we have modified our expressions in the paper.

---

### Meta-Review · Area_Chair1 · 2018-12-15
**Concerns with writing and technical novelty**

**Confidence:** 5
**Recommendation:** Reject

**Metareview:**

Reviewers have concerns about poor writing of the paper, lack of technical novelty, and the methodology taken by the paper not being very principled.